# Potential Biochemical Markers and Radiomorphometric Indices as Predictors of Reduced Bone Mass in Patients with Congenital Hemophilia

**DOI:** 10.3390/jcm11123391

**Published:** 2022-06-13

**Authors:** Sylwia Czajkowska, Joanna Rupa-Matysek, Ewelina Wojtasińska, Kacper Nijakowski, Lidia Gil, Anna Surdacka, Tomasz Kulczyk

**Affiliations:** 1Department of Conservative Dentistry and Endodontics, Poznan University of Medical Sciences, 61-701 Poznan, Poland; sylwia.budnik@gmail.com (S.C.); kacpernijakowski@ump.edu.pl (K.N.); annasurd@ump.edu.pl (A.S.); 2Department of Hematology and Bone Marrow Transplantation, Poznan University of Medical Sciences, 67-701 Poznan, Poland; j.rupamatysek@ump.edu.pl (J.R.-M.); ewelina.wojtasinska@skpp.edu.pl (E.W.); lidia.gil@skpp.edu.pl (L.G.); 3Department of Diagnostics, Poznan University of Medical Sciences, 67-701 Poznan, Poland

**Keywords:** hemophilia A, hemophilia B, X-rays, osteoporosis, extracellular traps

## Abstract

Background: The study aimed to evaluate radiomorphometric indices derived from panoramic X-rays and selected blood markers of bone turnover and neutrophil extracellular traps, with a view to identifying hemophilic patients at risk of developing osteoporosis. Methods: The study consisted of 50 adult men with hemophilia A and B (mild, moderate, and severe). The control group consisted of 25 healthy adult men. In both groups, blood samples were collected to determine concentrations of citrullinated histone H3 (CH3) and osteocalcin (BGLAP) with ELISA tests, and panoramic X-rays were obtained. Images were imported into AudaXCeph software to calculate two radiomorphometric indices: mental index (MI) and panoramic mandibular index (PMI). Concentrations of BGLAP and CH3 were compared with MI and PMI values in patients with and without hemophilia. Results: There were statistically significant differences in BGLAP, CH3, and PMI between the study and the control group (*p* < 0.05). Multivariate logistic regression analysis showed a predictive value for PMI, BGLAP, and CH3.The ROC curve with cutoff point (Youden index) at 0.40—PMI was calculated. No correlation was observed for the PMI index in any particular subgroup of patients. No correlation between MI and BGLAP/CH3 was observed. Conclusions: Simultaneous use of PMI value and BGLAP and CH3 levels may allow the identification of patients with hemophilia who requirea detailed diagnosis of osteoporosis with DXA.

## 1. Introduction

Hemophilia is a hereditary bleeding disorder marked by delayed blood clotting with prolonged or excessive bleeding caused by a deficiency of clotting factors.

Hemophilia can be classified as either hemophilia A (HA) for factor VIII deficiency (FVIII) or hemophilia B (HB) for factor IX deficiency (FIX). In HA and HB, the level of deficient clotting factor VIII (FVIII) and IX (FIX), respectively, correlates with the severity of symptoms, and a distinction is therefore made between mild (more than 5% factor activity), moderate (1–5% factor activity), and severe hemophilia (less than 1% factor activity, which corresponds to <0.01 IU/mL). Both HA and HB are inherited in a recessive manner and linked to the X chromosome; that is why it predominantly affects males (females can be affected in some rare cases of compound homozygosity, X chromosome loss or skewed X-inactivation) [1] with the incidence estimated as 1 in 4000 to 1 in 5000 live male births for hemophilia A and approximately 1 in 15,000 to 1 in 30,000 live male births for hemophilia B [2]. Hemophilia A is due to a variation in the F9 gene located on the most distal band (Xq28) of the long arm of the X chromosome which includes inversions, point mutations (missense and nonsense), small deletions and insertions, large deletions, and splice-site mutations, with intron 22 rearrangements (typically inversion) found in 40–45% of cases [3]. Mutations in the F9 gene that lead to qualitative and/or quantitative deficiencies are highly heterogenous due to missense variants (47% of variants), deletions, duplications, insertions, splice-site variants, and nonsense variants [4]. In patients with hemophilia, the imbalance of bone metabolism is persistent and can cause osteoporosis. Wallny et al. studied 62 male patients with severe hemophilia Awith a median age of 41 years and found reduced bone marrow density in 43.5% and osteoporosis in 25% of patients with HA [5]. According to Mansouritorghabeh et al. osteoporosis occurs in 37% of patients with hemophilia B [6]. Patients with hemophilia, with no or poor access to optimal care for joint and muscle bleedings, are more prone to developing reduced bone marrow density and osteoporosis, and the number of patients affected and severity of hemophilic arthropathy are associated with lower bone marrow density [5]. Additional risk factors for reduced bone density among patients with hemophilia were chronic hepatitis C, low BMI, and age [7]. Animal studies reveal a sexual dimorphism in the mechanism driving bone loss in males and females in F8 total knockout mice, consisting of a decline in bone formation in male mice but increased bone resorption in female mice [8]. Moreover, physical activity plays an important role in the prevention of osteopenia/osteoporosis in patients with hemophilia [9]. An association has been demonstrated between FVIII and FIX and decreased bone density in hemophilia. There are at least two known pathways by which FVIII/FIX can affect the skeletal system: osteoclast activity (cytokine-mediated) and mitogenic osteoblast activity (thrombin-mediated) [10]. The literature postulates a possible indirect mechanism mediated by the RANK-RANKL pathway and/or the Wnt/ β catenin pathway with osteoporosis development [11]. Moreover, reduced bone density in hemophilia may also be related to indirect factors such asdecreased activity due to fear of injury or the development of hemophilic arthropathy.

A hallmark of osteoporosis is a decrease in bone mineral density (BMD) and impaired bone microarchitecture, and consequently an increased risk of bone fractures. Techniques for diagnosing osteoporosis, such as Dual-energy X-ray absorptiometry (DXA), are limited and are not routinely used to detect the disease in patients with hemophilia. Finding a predictive factor for the development of osteoporosis and alternative methods of bone mineral density testing in the group of patients with hemophilia seems to be particularly important. Appropriate screening methods for osteoporosis in a group of patients with hemophilia could speed up the detection of osteopenia and osteoporosis and identify patients requiring urgent diagnosis using more sensitive methods.

Dentists are the primary-care physicians for many patients, particularlythose with hemophilia, because such patients may have a higher risk of developing caries [12]. One element of a typical dental examination is a panoramic X-ray image in which some dental and bone-involved problems can be visualized. So far, panoramic X-ray images have been successfully used in identifying patients with low bone mass [13,14,15,16,17], primarily postmenopausal women. The use of panoramic X-ray images as a screening tool was based on the particular types of measurements that could be performed in the mandibular region of a bone to calculate the so-called radiomorphometric indices. It has been shown that specific indices values correlate with low bone mass, and researchers have suggested detailed diagnostics for osteoporosis by means of sensitive methods such as DXA.

To our knowledge, there are no studies evaluating those radiomorphometric indices in patients with hemophilia as a tool to identify patients with low bone mass.

The aim of this study was to calculate panoramic X-ray–derived radiomorphometric indices in patients with hemophilia A and B and to compare them with selected hemophilic markers determined from the patient′s blood samples.

The study used osteocalcin (BGLA) as a marker of bone turnover; this is used in medical diagnostics to identify patients with low bone mass. The studies of Christoforidis et al. confirmed the usefulness of BGLA and reported an increase in BGLA concentration in the serum of patients with hemophilia [18,19]. Recently, there have also been claims about the possible influence of neutrophilic traps on the development of complications in hemophilic arthropathy that may be indirectly related to bone loss [20]. Due to the latest reports by Kamiński et al. on the influence of the formation of extracellular neutrophil traps on the body, we decided to include the neutrophil trap marker (citrullinated histone H3 (CH3)) [20].

## 2. Materials and Methods

### 2.1. Study Participants

The study group consisted of 50 men aged 19 to 65 (with a median age of 36.5) treated at the Department of Hematology and Bone Marrow Transplantation (DHBM) of PUMS and the Department of Conservative Dentistry and Endodontics of PUMS. Based on clinical evaluation and available medical documentation, patients were divided into hemophilia A and B subgroups, with the severity of hemophilia expressed as mild, moderate, or severe. An additional variable was the type of applied anti-bleeding therapy (on-demand vs. secondary prophylactic).

The control group consisted of 25 healthy male patients aged 21 to 63 treated for different reasons at the Department of Clinical Dentistry of PUMS. Patients younger than 18 years of age were excluded from the control group. Also, patients with a known history of factors that can influence or reduce bone mass, such as steroid therapy, chronic renal failure, prolonged immobilization, and parathyroid and thyroid disorders, were excluded from the control group. The data about the study group and the control group are presented in Table 1.

The study was conducted under the criteria of the Helsinki Declaration, and prior to the study, the patients′ written consent to participate in the study and the consent of the local Bioethics Committee were obtained (Resolution No. 628/20 with amendments approved by Resolution No. 210/21).

The research was performed between September 2020 and January 2022.

### 2.2. Radiological Study Protocol

For both the study and the control group, panoramic X-ray images were obtained as a part of the dental status evaluation (Figure 1). All the panoramic X-rays were taken with the same X-ray unit (Vistapano S, Durr Dental, Bietigheim-Bissingen, Germany) using the S-Pan proprietary manufacturer technology of automatic image creation from a large number of parallel layers. The X-rays were stored in the DICOM (Digital Imaging and Communications in Medicine) format for further evaluation of two indices, namely MI (Mental Index) and PMI (panoramic mandibular index), by means of the method proposed by Benson et al. [21]. The MI refers to the thickness of the mandibular cortical bone at the level of the mental foramen, while the PMI is calculated as the ratio of the cortical bone at the level of mental foramen to the distance between the inferior boundary of the mental foramen and the inferior border of the mandible (Figure 2). The MI and PMI were calculated employing AudaXCeph (Audax d.o.o., Ljubljana, Slovenia) software. This software can read DICOM medical images and allows for the creation of customized two-dimensional measurements. The analysis of indices was carried out by a single researcher. Three months after completing the measurements, 10 randomly selected panoramic X-rays were analyzed again to calculate the intraclass correlation coefficient (ICC). The obtained value of 0.9392 suggested strong agreement in the assessment.

### 2.3. Biochemical Study Protocol

For every participant, 10 mL of peripheral blood was collected (in vacuum tubes containing the anticoagulant–EDTA) to determine the concentrations of citrullinated histone H3 (CH3) and bone gamma–carboxyglutamate protein–osteocalcin (BGLAP). To avoid the daily deviations of the concentrations of the tested markers, blood samples were preserved in the morning and stored at −70 to −80 °C before performing the analysis. The levels of CH3 and BGLAP were determined by commercial ELISA tests (Shanghai Sunred Biological Technology Co., Shanghai, China).

### 2.4. Medical Documentation

The available medical records provided key information on the type of hemophilia (A, B), severity (mild, moderate, severe), treatment, onset of hemophilic arthropathy, and patient′s age.

### 2.5. Study Assumptions

The aim of the study was to compare the values of BGLAP and CH3 levels as well as MI and PMI values in patients with and without hemophilia and to find any correlation between these indicators depending on the course of the disease, which could contribute to faster identification of patients with hemophilia who require detailed diagnostics of osteoporosis (DXA).

### 2.6. Statistical Analysis

Data were analyzed using Statistica 13.3 (StatSoft, Cracow, Poland). The significance level was set at α = 0.05. Due to noncompliance with the normal distribution (Shapiro–Wilk test) for the BGLAP and CH3 concentrations and the PMI and MI index values, the results were presented in box plots as medians and quartile ranges (the upper and lower quartile corresponding to 25th and the 75th percentiles, respectively). 

The variables of the two independent groups were compared using the nonparametric Mann–Whitney U test. The correlations between the variables were assessed using the Spearman correlation coefficient. A predictive assessment of the PMI value was performed using ROC analysis, and the cutoff values were determined according to the Youden index. All factors were then tested in stepwise regression analysis, progressing with V-fold validation.

## 3. Results

### 3.1. Sample Characteristics

Table 1 presents the characteristics of the study and the control group.

In the study group, additional data about the type of hemophilia (A or B), severity of the disease (mild, moderate, or severe) and the applied anti-bleeding therapy (on-demand vs. secondary prophylactic) are included. These data were retrieved from the medical files of patients from the Department of Hematology and Bone Marrow Transplantation (DHBM).

### 3.2. Biochemical and Radiological Analysis

There were statistically significant differences in the BGLAP concentrations, CH3 concentration, and PMI value between the study and the control group (Table 2, Figure 3, Figure 4 and Figure 5). However, no correlation was found for PMI in any particular subgroups of patients divided into type, severity, or type of therapy.

Also, no correlation was found for PMI in the younger patients. There was no correlation between the severity of hemophilia (severe versus moderate and mild) and the concentrations of BGLAP, CH3, and PMI. Moreover, no relationship was found between the occurrence of arthropathy and the PMI value (*p* > 0.05).

Figure 6 shows the ROC curve with the proposed cutoff point of 0.40 (Youden index) made for the predictive assessment of the PMI value AUC = 0.697, SE = 0.065, *p*-Value = 0.0024. The analysis confirmed that the PMI value calculated, based on the panoramic X-ray, shows predictive values; higher values indicated a greater probability of hemophilia.

### 3.3. Multivariate Logistic Regression Analysis

Logistic univariate regression analysis showed a prognostic value for PMI, BGLAP, CH3 (*p*-Value 0.023 for PMI, 0.002 for BGLAP, and <0.001 for CH3), which allowed for multivariate analysis. The results of stepwise regression analysis with V-fold validation (the obtained ROC curves for the learning and validation samples were compared) are presented in Table 3, Figure 7. The Hosmer–Lemeshow test showed a good fit of the model to the data (*p*-Value 0.975).

## 4. Discussion

The increased availability of clotting factor concentrates and the introduction of prophylaxis in the 1990s have led to an extension of the life expectancy of patients with hemophilia.

The role of comorbidities and the negative influence of hemophilia on bone mineral density have more often been demonstrated, and attention has been drawn to the risk of developing secondary osteoporosis [22]. Numerous studies have shown that adult patients with hemophilia are at increased risk of developing secondary osteoporosis [23,24,25,26,27,28]. However, due to many factors that predispose to decreased bone mass, the pathogenesis of osteoporosis in hemophilia has not been fully elucidated [29]. Iorio et al. [24] performed a meta-analysis of seven case-control studies assessing bone mineral density with DXA, including one study of 100 adult patients and another of 111 pediatric patients, and concluded that patients with severe hemophilia are particularly at risk of developing osteoporosis.

The high risk of development of bone disease in patients with moderate and severe hemophilia was confirmed by Gerstner et al. [23], and also by KiperUnal et al. [25]. The first-mentioned study was conducted in Arizona (USA)with a group of 30 adult patients with coagulation factor VIII or IX levels <5% (moderate hemophilia) and <1% (severe hemophilia) [23]. According to the researchers, osteopenia occurred in 43% of the patients and osteoporosis (confirmed by DXA) occurred in 25% of the patients. The authors excluded from the study any patients who had taken the agent prophylactically before the age of 10, which could hypothetically affect bone turnover. In the study by KiperUnal et al. [25], carried out in Turkey on 49 patients aged 20–60 years with moderate and severe hemophilia, an increased incidence of low BMD among patients with factor VIII and IX deficiency [25] was reported. Lower BMD than the reference age was demonstrated in over a third of hemophilic patients (34.8%) under the age of 50. However, in the case of patients over 50 years of age, as many as 66.6% of patients were diagnosed with osteoporosis. In order to assess osteoporosis and osteopenia, researchers from Turkey also performed DXA (scan of the lumbar spine, femoral neck, and hip) and used the WHO classification system, which makes the study objective and reliable [25].

In a study conducted at the Medical University of Atlanta (United States) on a group of 88 patients with hemophilia A and B, the results showed that the increased incidence of osteoporosis was not limited to patients with severe and moderate hemophilia but could also affect patients with mild hemophilia (>5% factor) [28]. In this study, BMD was measured by DXA (hip bone, spine, and right and left femoral necks), and the advantage of the analysis was the collection of a large and racially diverse study group. Based on the obtained results, Kempton et al. showed that in hemophilia patients under 50 years of age, the increase in the incidence of osteoporosis correlates with the severity of hemorrhagic diathesis. However, at the same time, researchers showed that osteoporosis appears as a comorbid disease in almost 40% of hemophilia patients over 50 years of age, with the distribution not changing significantly depending on the concentration of the deficient coagulation factor. Kempton et al. documented a correlation between the concentration of the deficient factor and the incidence of osteoporosis in patients under 50 years of age. However, our study group did not correlate the patient′s age and the PMI value.

Considering the results of the studies mentioned above and the increased risk of developing osteoporosis, primarily in patients with severe hemophilia, we have focused mainly on patients with a factor level <1%. However, we decided not to exclude patients with mild and moderate hemophilia from the study, in whom osteoporosis may also be a comorbid disease.

Our study used osteocalcin as an important marker of bone turnover—a protein responsible for bone mineralization which also plays a vital role in medical diagnostics in identifying patients at high risk of bone fractures. The use of osteocalcin was also supported by studies confirming an increase in the concentration of this marker in the serum of patients with hemophilia [18,19]. Based on a report pointing to the possible impact of extracellular neutrophil traps on the development of complications in hemophilia, we decided to include citrullinated histone H3 in the study [20].

We would like to emphasize that our goal was not to unambiguously diagnose osteoporosis but to identify patients with hemophilia who needed an urgent diagnosis for bone loss. The relationship between osteoporosis and bone loss within the structures of the oral cavity began to be noticed only in the second half of the 1990s when the relationship between periodontal disease and spinal osteoporosis was described [30,31]. The index used in the present study—PMI—was proposed in 1991 by Benson et al. and is based on the assessment of the thickness of the compact bone below the mental foramen as there are no masticatory muscles in this region [21]. When creating the index, researchers relied on previous findings that the distance between the foramen and the lower jaw remains relatively constant throughout life [32]. Later studies seem to confirm the possible usefulness of PMI in identifying patients with low bone mass [33,34,35]. Also, some semi-automatic methods of determining the indices were proposed [36]. PMI is not the only factor taken into account in the craniofacial region; there are others. So far, the most extensive study using several existing methods of mandibular bone density classification is the multi-center project OSTEODENT [37,38,39,40,41,42]. The research carried out as part of the project did not indicate a method of assessing a panoramic X-ray that could effectively replace DXA in the diagnosis of osteopenia and osteoporosis; however, it highlighted the potential of dental radiology in identifying patients with low bone mass and the need for further research in this area.

We believe that both hematologists and dentists can use the results of our research.

Our study showed a statistically significant difference in the PMI values in the test group and control group (with a cutoff point of 0.40), which seems to confirm the higher incidence of osteoporosis and osteopenia in patients with hemophilia A and B. Thanks to clinical data and the results of blood tests, we were able to compare the PMI values with the marker of bone turnover and neutrophil traps (NETs). Multivariate analysis of variance showed that the comparison of PMI with the concentration of osteocalcin and citrullinated histone H3 makes it possible to predict whether the patient is deficient in factor VIII/IX.The advantage of this approach to classical FVIII/FIX deficiency may be the possibility of using the mentioned correlation for diagnostic purposes when it is not possible (e.g., for technical reasons) to measure factor VIII and IX levels in the patient. The demonstration of the PMI/CH3/BGLAP correlation may be an indication to perform genetic tests for hemophilia A and B. This situation may primarily concern patients undergoing hematological diagnostics in whom a spontaneous mutation occurred and where family history did not indicate hemophilia. Moreover, the aforementioned PMI/CH3/BGLAP correlation may be an indication for diagnosis due to excessive and spontaneous bleeding in women suffering with hemophilia or carriers who have not yet been diagnosed.

Dentists, particularly periodontists who most often treat patients with hemophilia, use panoramic images during treatment. Determination of the PMI index in patients with hemophilia can be a means of preliminary and rapid identification of individuals with reduced bone mass, who should be referred for detailed diagnostics using precise methods such as DXA.

Another potential concern may be when the patient has increased bleeding in the oral cavity that does not correlate with gingival inflammation. In such a situation, the PMI calculated based on a panoramic X-ray could indicate the most likely diagnostic direction—hematological or periodontal.

Osteocalcin is a protein responsible for bone mineralization, and PMI is a valuable parameter in the identification of patients with low bone mass [43], which suggests that statistically significant differences in the values of these parameters in patients with hemophilia A and B, compared to those in the control group, indicate that they were highly likely to be associated with osteopenia/osteoporosis. The effect of neutrophilic trap formation on bone health is unknown, but interaction with the variables mentioned above suggests a possible negative effect of NETs on bone mass. A panoramic X-ray–derived index, in combination with laboratory tests (BGLAP, CH3), can be a tool for identifying patients with hemophilia, especially those at risk of bone loss. DXA testing is still limited, and having an additional test that can correctly identify patients at risk may accelerate the diagnosis of osteoporosis in this group of patients and thus positively affect patients’ quality of life. Due to financial limitations, we could not correlate the obtained measurements with DXA values. However, we hope to extend this research in the future within the “National Program of Treatment of Patients with Hemophilia and Related Hemorrhagic Diathesis for adult patients,” which is performed in our institution.

## 5. Conclusions

The study demonstrates the usefulness of PMI in identifying hemophilic patients at risk of reduced bone mass. The reliability of the measurement is enhanced by a comparison of the obtained result with the concentration of osteocalcin and citrullinated histone H3. The simultaneous use of these three markers may enable the identification of patients with hemophilia who require a more detailed diagnosis of osteoporosis. The results of this study may be introduced into clinical practice quickly, as a panoramic radiograph is a widely used diagnostic tool in dentistry and PMI calculation by a properly trained dentist is not a time-consuming procedure.

## Figures and Tables

**Figure 1 jcm-11-03391-f001:**
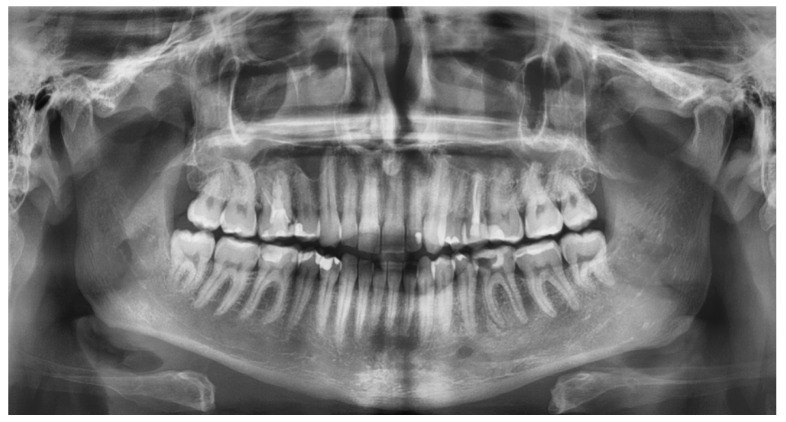
An exemplary panoramic radiograph used in this study made on Vistapano S, Durr Dental Germany (This X-ray presents a patient from the study group).

**Figure 2 jcm-11-03391-f002:**
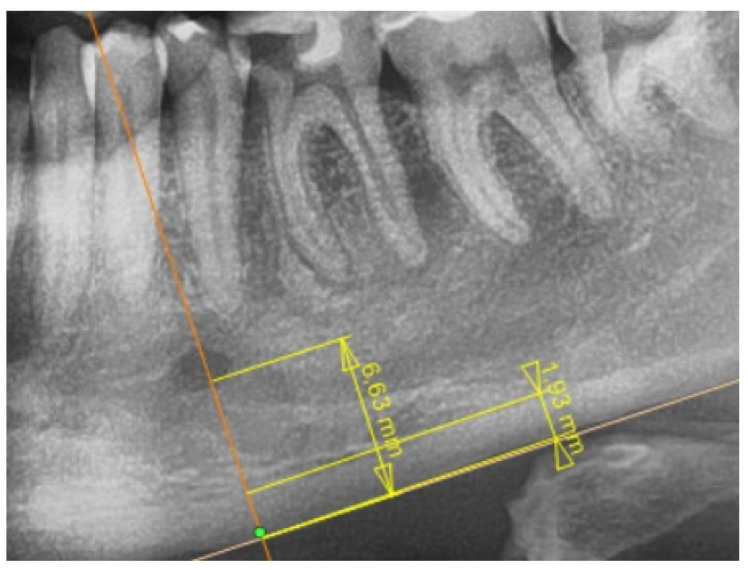
Cropped panoramic radiograph (from a Figure 1) of the mental foramen region with tracing and measurements performed with the AudaXCeph software.

**Figure 3 jcm-11-03391-f003:**
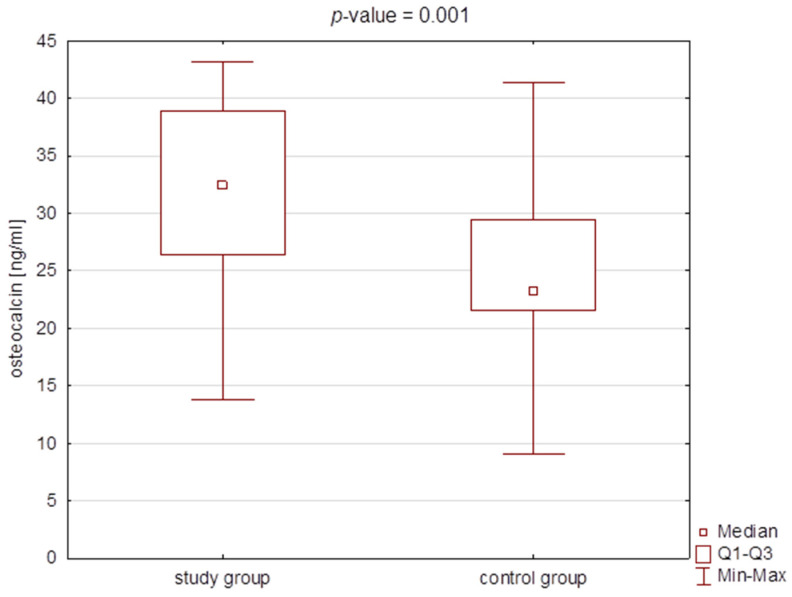
Box plot for osteocalcin (BGLAP) concentration in the study group and the control group.Q1–lower quartile; Q3–upper quartile.

**Figure 4 jcm-11-03391-f004:**
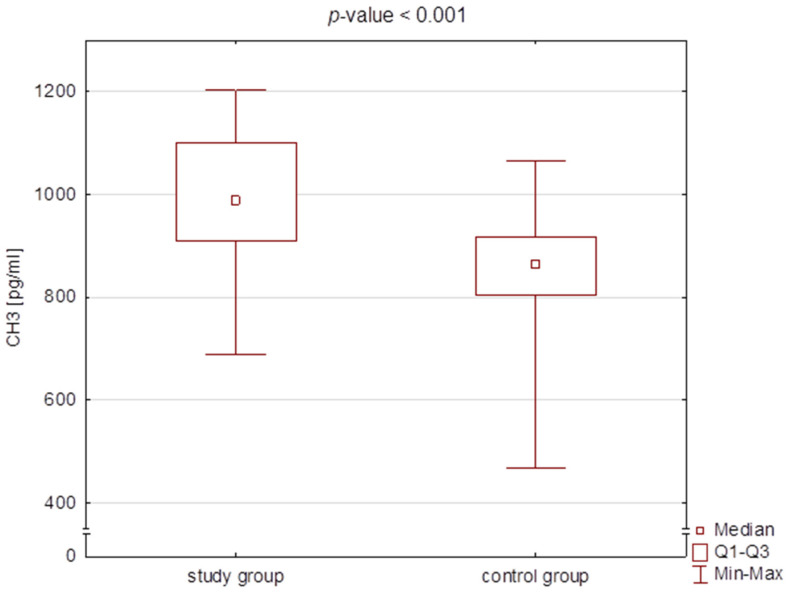
Box plot for citrullinated histone H3 (CH3) concentration in the study group and the control group.

**Figure 5 jcm-11-03391-f005:**
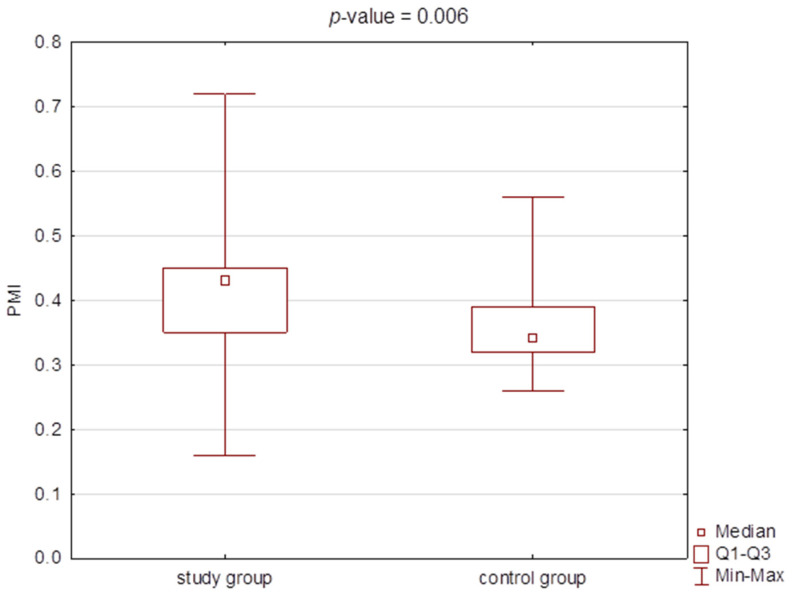
Box plot forpanoramic mandibular index (PMI) values in the study group and the control group.

**Figure 6 jcm-11-03391-f006:**
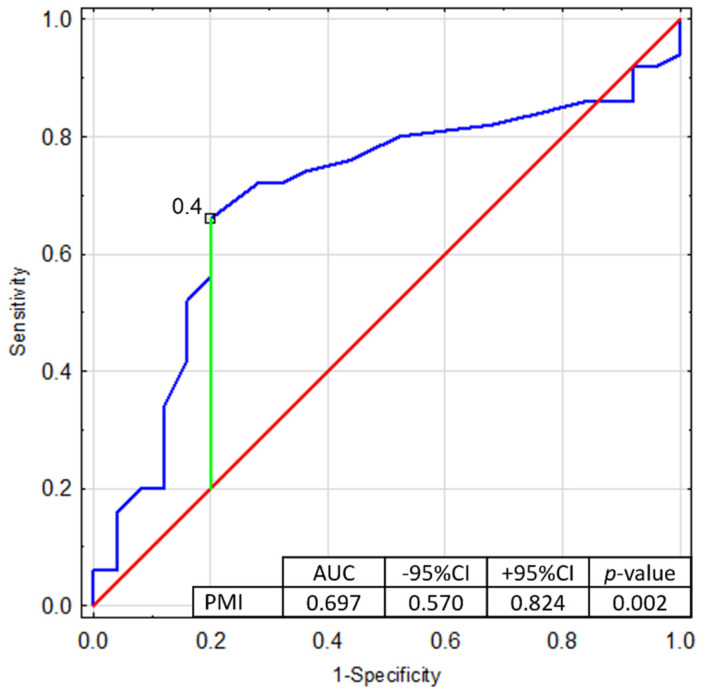
Receiver operating characteristic curve for PMI. AUC—Area Under Curve, SE—standard error, CI—confidence interval.

**Figure 7 jcm-11-03391-f007:**
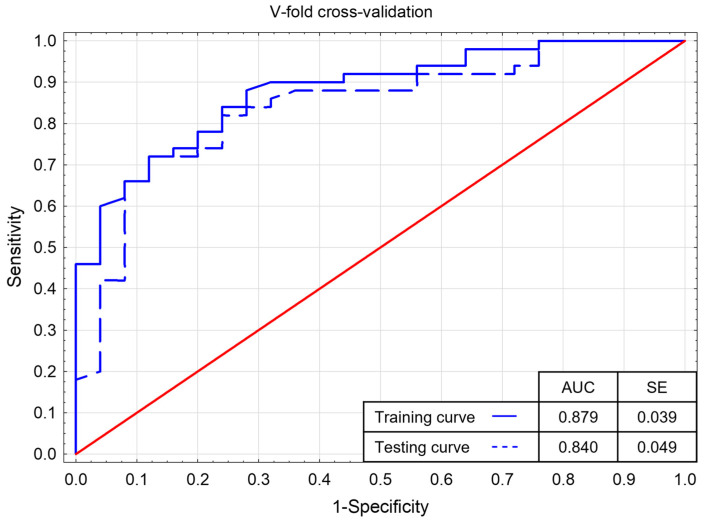
V-fold cross-validation—learning ROC curves for the logistic regression model. AUC—Area Under Curve, SE—standard error.

**Table 1 jcm-11-03391-t001:** Characteristics of the study and control groups.

		*n* (%)	Age (Range)
**Control Group**		25	28 (21–63)
**Study Group**		50	36.5 (19–65)
**Type of hemophilia**	A	40 (80%)	36.0 (19–65)
	B	10 (20%)	41.5 (26–56)
**Routine management**	On demand	22 (44%	33.5 (20–65)
	Secondary prophylactic	28 (56%)	38.5 (19–62)
**Severity of hemophilia**	Severe	35 (70%)	40.0 (19–62)
	Moderate	7 (14%)	32.0 (26–56)
	Mild	8 (16%)	24.5 (20–65)

**Table 2 jcm-11-03391-t002:** Comparison of the concentrations of osteocalcin (BGLAP), citrullinated histone H3 (CH3), and the values of PMI (panoramic mandibular index) and MI (mental index) between the study group and the control group (median and Q1–Q3).

	Study Group *n* = 50		Control Group *n* = 25		*p*-Value
	M [Q1–Q3]	Min–Max	M [Q1–Q3]	Min–Max	
BGLAP [ng/mL]	32.46 [26.41–38.85]	9.03–41.33	23.25 [21.54–29.44]	13.83–43.21	0.001 *
CH3 [pg/mL]	987.3 [909.2–1101.3]	468.0–1064.8	863.6 [803.2–917.8]	687.5–1203.9	<0.001 *
PMI	0.43 [0.35–0.45]	0.16–0.72	0.34 [0.32–0.39]	0.26–0.56	0.006 *
MI	2.48 [2.28–2.69]	1.32–3.68	2.37 [2.03–2.59]	1.77–2.93	0.167

* *p*-value < 0.05 for the Mann–Whitney U test.

**Table 3 jcm-11-03391-t003:** Parameters of predictors incorporated into the logistic regression model. SE—standard error.

	Β	SE	Wald Stat.	*p*-Value	Odds Ratio	Confidence OR −95%	Confidence OR 95%
Intercept	−16.947	4.365	15.077	<0.001 *			
CH3 [pg/mL]	0.012	0.004	10.602	0.001 *	1.012	1.005	1.020
BGLAP [ng/mL]	0.130	0.045	8.469	0.004 *	1.139	1.043	1.243
PMI	6.988	3.471	4.054	0.044 *	1083.675	1.204	975,220.994

* Statistical significance *p*-value < 0.05.

## Data Availability

Due to the nature of this research, participants of this study did not agree for their data to be shared publicly, so supporting data are not available.

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
