# Peer review of "Potential Biochemical Markers and Radiomorphometric Indices as Predictors of Reduced Bone Mass in Patients with Congenital Hemophilia"

_jcm, 2022, doi:10.3390/jcm11123391_

Round 1

Reviewer 1 Report

Review of Potential biochemical markers and radio-morphometric indices as predictors of reduced bone mass in patients with congenital hemophilia by Sylwia Czajkowska et al 2022

Sylwia Czajkowska and colleagues present a useful method for the diagnosis of bone loss in association with hemophilia however some questions need to be approached to increase quality and clarify the manuscript. Questions and suggestions follow below:

In abstract osteocalcin abbreviation is one of the abbreviations commonly accepted, the official gene and protein name is bone gamma-carboxyglutamate protein with the official abbreviation being BGLAP. It is acceptable when osteocalcin in mentioned the abbreviation be OC and not OT

In addition, authors mention that PMI corresponds to Mental index, but bibliography refers that PMI is defined as panoramic mandibular index which allows a better understanding what the authors are talking about in stead of mental index….

The authors mention that “Hemophilia is a hereditary bleeding disorder marked by delayed blood clotting with prolonged or excessive bleeding caused by a deficiency of clotting factors”. However, attending that is a hereditary disease well characterized authors should review main mutations and epidemiology of the disease, namely the fact that Hemophilia is caused by a lack of antihemophilic factor(s), for example, factor VIII (FVIII; hemophilia A) and factor IX (FIX; hemophilia B). In addition, authors did not review which are the known mechanisms associated which favors bone resorption in opposition to bone formation, already revised in literature.

In methods the authors did not explain nor present any citation to validate their approach to use as molecular markers for bone metabolism citrullinated histone H3 and osteocalcin. Osteocalcin is a common marker as bone formation but citrullinated histone H3 is not. Please explain why were used citrullinated histone H3 and osteocalcin as markers and provide bibliography to support that or preliminary results. This is only introduced in discussion “ Based on a report pointing to the possible impact of 233 extracellular neutrophil traps on the development of complications in hemophilia, we decided to include citrullinated histone H3 in the study [20].” Must be contextualized early in the manuscript to contribute for results interpretation.

The authors present study assumptions “ The aim of the study was to compare the values of OT and CH3 levels as well as MI and PMI values in patients with and without haemophilia and to find any correlation between these indicators depending on the course of the disease” but they not explain why is important within the context of the disease and progression to osteoporosis to obtain this correlation. Please clarify

An example of the lack of contextualization is the fact that authors mention in results that there are groups of patients with type of hemophila ( A or B) but the authors did not contextualize hemophilia A and B in introduction.

The data presented in table 2 would benefit if authors could add confidence intervals at 95%, variation of the groups are more important than max and min values under a context of comparison between different groups and because they are already expressed in box plot graphics.

Authors in discussion mention that “The multivariate analysis of variance showed that the comparison of PMI with the concentration of osteocalcin and citrullinated histone H3 makes it possible to predict whether the patient is deficient in factor VIII / IX.” Can the authors discuss what is the advantage of this approach to classic methods for deficiency in factors VIII and IX?

Author Response

We wish to resend our review article, "Potential biochemical markers and radio-morphometric indices as predictors of reduced bone mass in patients with congenital hemophilia," Thank you very much for the suggestions and all the constructive and valuable comments, which have allowed us to improve the quality of the paper. Our point-by-point responses to them are given below. Changes are marked in the revised manuscript using the "Track Changes" function in Microsoft Word.
All the suggestions made by the Reviewers have been implemented in the current version of the paper, and the responses to the comments raised by the Reviewers are attached below.
We are very grateful for all of your comments which have allowed us to improve the quality of the paper.

Review 1:

Comments:In abstract osteocalcin abbreviation is one of the abbreviations commonly accepted, the official gene and protein name is bone gamma-carboxyglutamate protein with the official abbreviation being BGLAP. It is acceptable when osteocalcin in mentioned the abbreviation be OC and not OT

Reply: Thank you very much for this comment. We followed the abbreviation used by the manufacturer of  the Elisa tests. Of course, we agree with the above statement, so we have corrected the abbreviation, OT, in the manuscript to the official abbreviation BGLAP. This is a valuable comment, and the use of the official abbreviation will make it easier to understand the text and allow other researchers to more easily find our paper.

Comments:In addition, authors mention that PMI corresponds to Mental index, but bibliography refers that PMI is defined as panoramic mandibular index which allows a better understanding what the authors are talking about in stead of mental index….

Reply: Thank you for this suggestion. The PMI definition has been corrected to be more explicit and independent of the MI.

Comments:The authors mention that "Hemophilia is a hereditary bleeding disorder marked by delayed blood clotting with prolonged or excessive bleeding caused by a deficiency of clotting factors". However, attending that is a hereditary disease well characterized authors should review main mutations and epidemiology of the disease, namely the fact that Hemophilia is caused by a lack of antihemophilic factor(s), for example, factor VIII (FVIII; hemophilia A) and factor IX (FIX; hemophilia B). In addition, authors did not review which are the known mechanisms associated which favors bone resorption in opposition to bone formation, already revised in literature.

Reply: Thank you for this suggestion. We have made changes to the introduction, which right now contain the following part:

“Hemophilia can be classified as either hemophilia A (HA) for factor VIII deficiency (FVIII) or hemophilia B (HB) for factor IX deficiency (FIX). In HA and HB, the level of deficient clotting factor VIII (FVIII) and IX (FIX), respectively, correlates with the severity of symptoms, and a distinction is therefore made between mild (more than 5% factor activity), moderate (1–5% factor activity), and severe hemophilia (less than 1% factor activity, which corresponds to <0.01 IU/mL). Both HA and HB are inherited in a recessive manner, linked to the X chromosome, that is why it predominantly affects males (females can be affected in some rare cases of compound homozygosity, X chromosome loss or skewed X-inactivation) with the incidence estimated as 1 in 4000 to 1 in 5000 live male births for hemophilia A and approximately 1 in 15000 to 1 in 30000 live male births for hemophilia B [2]. Hemophilia A is due to variation in the F9 gene located on the most distal band (Xq28) of the long arm of the X chromosome which includes inversions, point mutations (missense and nonsense), small deletions and insertions, large deletions, and splice-site mutations with intron 22 rearrangements (typically inversion) most often found in 40-45% of cases. Mutations in the F9 gene that lead to qualitative and/or quantitative deficiencies are highly heterogenous due to missense variants (47 percent of variants), deletions, duplications, insertions, splice-site variants and nonsense variants. In patients with hemophilia, the imbalance of bone metabolism is persistent and can cause osteoporosis. Wallny et al. studied 62 male patients with severe hemophilia A, with a median age of 41 years, and found reduced bone marrow density in 43.5% and osteoporosis in 25% of patients with HA. According to Mansouritorghabeh et al. osteoporosis occurs in 37% of patients with hemophilia B. Patients with hemophilia, with no or poor access to optimal care for joint and muscle bleedings, are more prone to developing reduced bone marrow density and osteoporosis, and the number of patients affected and severity of hemophilic arthropathy are associated with lower bone marrow density. Additional risk factors for reduced bone density among patients with hemophilia were chronic hepatitis C, low BMI and age. Animal studies reveal a sexual dimorphism in the mechanism driving bone loss in males and females in F8 total knockout mice consisting of a decline in bone formation in male mice but increased bone resorption in female mice. Also physical activity plays an important role in the prevention of osteopenia/osteoporosis in patients with hemophilia. An association has been demonstrated between FVIII and FIX and decreased bone density in hemophilia. There are at least two known pathways by which FVIII / FIX can affect the skeletal system: osteoclast activity (cytokine-mediated) and mitogenic osteoblast activity (thrombin-mediated). Literature postulates a possible indirect mechanism mediated by the RANK-RANKL pathway and/or Wnt/ β catenin pathway with osteoporosis development. Moreover, reduced bone density in hemophilia may also be related to indirect factors such as, for example, decreased activity due to fear of injury or the development of hemophilic arthropathy.”

Comments:In methods the authors did not explain nor present any citation to validate their approach to use as molecular markers for bone metabolism citrullinated histone H3 and osteocalcin. Osteocalcin is a common marker as bone formation but citrullinated histone H3 is not. Please explain why were used citrullinated histone H3 and osteocalcin as markers and provide bibliography to support that or preliminary results. This is only introduced in discussion "Based on a report pointing to the possible impact of 233 extracellular neutrophil traps on the development of complications in hemophilia, we decided to include citrullinated histone H3 in the study [20]." Must be contextualized early in the manuscript to contribute for results interpretation.

Reply: Thank you very much for your important suggestion. In the introduction, we decided to explain why the study used osteocalcin as a marker of bone turnover and a marker of the neutrophilic trap - citrullinated histone H3. The study used osteocalcin (BGLA) as a marker of bone turnover, which is used in medical diagnostics to identify patients with low bone mass. The studies of Christoforidis et al. confirmed the usefulness of BGLA, they reported an increase in BGLA concentration in the serum of patients with hemophilia. Recently, there have also been claims about the possible influence of neutrophilic traps on the development of complications in hemophilic - arthropathy that may be indirectly related to bone loss[20]. Due to the latest reports by Kamiński et al. on the influence of the formation of extracellular neutrophil traps on the body, we decided to include the neutrophil trap marker (citrullinated histone H3 (CH3)).

Comments: The authors present study assumptions "The aim of the study was to compare the values of OT and CH3 levels as well as MI and PMI values in patients with and without hemophilia and to find any correlation between these indicators depending on the course of the disease" but they not explain why is important within the context of the disease and progression to osteoporosis to obtain this correlation. Please clarify

Reply: As suggested, we explained the purpose of looking for correlations between MI, PMI, CH3 and BGLA values. The aim of the study was to compare the values of BGLAP and CH3 levels as well as MI and PMI values in patients with and without hemophilia and to find any correlation between these indicators depending on the course of the disease, which could contribute to faster identification of patients with hemophilia who require detailed diagnostics of osteoporosis (DXA). A detailed explanation of the significance of the analyzed correlation was also described in the conclusions. The study demonstrates the usefulness of the PMI in identifying hemophilic patients at risk of reduced bone mass. The reliability of the measurement is enhanced by a comparison of the obtained result with the concentration of osteocalcin and citrullinated histone H3. The simultaneous use of these three markers may enable the identification of patients with hemophilia who require a more detailed diagnosis of osteoporosis. The results of this study may be introduced into clinical practice quickly as a panoramic radiograph is a widely used diagnostic tool in dentistry, and PMI calculation by a properly trained dentist is not a time-consuming procedure.

Comments: An example of the lack of contextualization is the fact that authors mention in results that there are groups of patients with type of hemophilia ( A or B) but the authors did not contextualize hemophilia A and B in introduction.

Reply: Thank you for your comment, which allowed us to add some important information in the introduction. Information about haemophilia’s variants has now been added into the introduction part of our manuscript.

Comments: The data presented in table 2 would benefit if authors could add confidence intervals at 95%, variation of the groups are more important than max and min values under a context of comparison between different groups and because they are already expressed in box plot graphics.

Reply: Thank you for this comment. Confidence intervals are usually given for variables with a normal distribution that did not occur in the study. For this reason, we decided to use medians and quartile ranges. For a better understanding of the results for non-statistic readers, we decided to duplicate (box plots, table 2) the results that the authors believed were relevant. However, we leave it to the editor's decision to remove the minimum and maximum values from table 2.

Comments: Authors in discussion mention that "The multivariate analysis of variance showed that the comparison of PMI with the concentration of osteocalcin and citrullinated histone H3 makes it possible to predict whether the patient is deficient in factor VIII / IX." Can the authors discuss what is the advantage of this approach to classic methods for deficiency in factors VIII and IX?

Reply: Thank you for this important question. Our study showed a statistically significant difference in the PMI values in the test group and control group (with a cut-off point of 0.40), which seems to confirm the higher incidence of osteoporosis and osteopenia in patients with hemophilia A and B. Thanks to clinical data and the results of blood tests we were able to compare of the PMI values with the marker of bone turnover and neutrophil traps (NETs). Multivariate analysis of variance showed that the comparison of PMI with the concentration of osteocalcin and citrullinated histone H3 makes it possible to predict whether the patient is deficient in factor VIII / IX. The advantage of this approach to classical FVIII/FIX deficiency may be the possibility of using the mentioned correlation for diagnostic purposes when it is not possible (e.g., for technical reasons) to measure factor VIII and IX levels in the patient. The demonstration of the PMI/CH3/BGLAP correlation may be an indication to perform genetic tests for hemophilia A and B. This situation may primarily concern patients undergoing hematological diagnostics in whom a spontaneous mutation occurred, and the family history did not indicate hemophilia. Moreover, the mentioned PMI/ CH3/ BGLAP correlation may be an indication of diagnosis due to excessive and spontaneous bleeding in women suffering with hemophilia or carriers who have not yet been diagnosed.

Dentists, particularly periodontists who most often treat patients with hemophilia, use panoramic images during treatment. Determination of the PMI index in patients with hemophilia can be a means of preliminary and rapid identification of individuals with reduced bone mass, who should be referred for detailed diagnostics using precise methods such as DXA.

Another potential concern may be when the patient has increased bleeding in the oral cavity that does not correlate with gingival inflammation. In such a situation, the PMI calculated based on a panoramic X-ray could indicate the most likely diagnostic direction - hematological or periodontal.

Reviewer 2 Report

Prognosis and life expectancy for patients with hemophilia have improved immensely over the last decades. Thanks to factor concentrates, affected patients can prevent major bleedings and live an almost normal life. However, nowadays we are faced with the accumulating evidence, that patients with hemophilia are prone to low bone mineral density, leading to osteopenia and potentially to osteoporosis. Diagnosis thereof usually consists of DXA measurements, which is often limited and not routinely used; robust and sensitive biochemical markers are still lacking. In the submitted article, the authors propose panoramic x-rays, performed by a dentist, in combination with the assessment of osteocalcin and citrullinated histone H3 as predictors of reduced bone mass in patients with congenital hemophilia.

The manuscript is written in a clear manner, and the statistics are technically sound. However, I have some major concerns regarding one of the used markers and the overall interpretation of the data.

Major comments:

  • I do wonder why the authors used citrullinated histone H3 (CH3) as a marker for osteoporosis? As they mentioned in the discussion, there is thus far only one report pointing to a possible impact of extracellular neutrophil traps on the development of joint injury in hemophilia. However, neither is this marker well approved yet, nor is it associated with bone mineralization or hemophilia in general.
  • The authors claim that the combination of PMI, together with laboratory testing of OT and CH3 allows the prediction of low bone mass in patients with hemophilia. I do wonder how the authors are able to draw this conclusion. As they write in the discussion, this may be rather a tool to identify patients with hemophilia, but more - is from my point of view – rather speculative. This is pronounced by the finding, that there is no difference in the concentrations of OT, CH3 and the PMI between the severity of hemophilia groups (mild vs severe). Here, DXA measurements and assessing the bone mineralization status of the study participants is lacking in order to draw any conclusions.
  • A study from 2015, namely ‘The evaluation of MCI, MI, PMI and GT on both genders with different age and dental status’ by Bozdag and Sener could show that dental status had a significant effect on the MI and PMI in males. I thus wonder if the authors have assessed the dental status in their study since this could be a confounding factor?
  • As you see on the long list of minor comments, I highly recommend careful proofreading of the manuscript.

Minor comments:

  • Why do the authors use different numbers although having the same affiliation (2-4)?
  • The authors use twice the spelling haemophilia instead of hemophilia (in the Abstract and in Section 2.4). Please change this in order to be consistent. The same is true for Hematology/Haematology.
  • In the Abstract several space characters are missing, e.g. lane 25: …ELISA testsand panoramic…, lane 30: …CH3.The ROC…, lane 34: …who requirea detailed.. Simultaneously, there may be some space characters too much (lane 32, 33).
  • Page 2, lane 50: …for most patients, particularly FOR those with hemophilia…
  • Page 2, lane 87: panoramic x-ray images and not x-rays images
  • Page 2, lane 89: there might be a space character too much between Vistapano and S
  • Page 2, lane 91: there might be a space character too much after: The x-rays were…
  • Page 2, lane 92/93: the authors mixed up the abbreviations for MI and PMI, respectively.
  • Page 2, lane 93: please add the according reference number after Benson et al.
  • Page 3: lane 99: there might be a space character too much after the point.
  • Page 3: lane 107: there might be a space character too much after Cropped.
  • Page 4: lane 127: please write 25th instead of 25.
  • Page 4: lane 129: it is called Mann-Whitney U test.
  • Table 1: there might by a space character too much before ‘Type of hemophilia’
  • Table 1: I guess ‘The Control Group’ should be written at the same height as ‘The Study Group’
  • Page 4: lane 143: the authors write PMI concentrations; please change the wording.
  • Page 4: lane 145: please only write PMI and not PMI index.
  • Page 4: Section 3.2.: At first, the authors write: No correlation was found for the PMI index in particular subgroups of patients divided into type, severity, or type of therapy. However, two sentences later they write: There was no correlation between the severity of hemophilia… and the PMI value. Isn’t this the same?
  • Page 4: lane 150: Until here, nowhere was mentioned, that the occurrence of arthropathy was also assessed.
  • Table 2: It should be ‘citrullinated histone H3 (CH3)’ instead of ‘citrullinated histone CH3’.
  • Table 2: PMI already stands for panoramic mandibular index, so please write PMI instead of PMI index.
  • Table 2: It is called Mann-Whitney U test and not Mann-Whitney test.
  • Figure 6: After the point a space character is missing.
  • Table 3: Please list the abbreviations used in the legend.
  • Table 3: Please check the β value for the intercept.
  • Figure 7: for me the caption is not really clear: for what stands the ‘re’? Please also provide information for the abbreviations to the reader.
  • Page 8, lane 208: Please add reference.
  • From my point of view, the discussion can be shortened. The first two paragraphs, listing the findings of low bone mineral density in patients with hemophilia, is too long, especially since those results were already summarized in excellent review articles.
  • The section ‘Author Contributions’ should be over-worked.
  • Section ‘Data Availability Statement’: Sentence written in double; delete one.

Author Response

We wish to resend our review article, "Potential biochemical markers and radio-morphometric indices as predictors of reduced bone mass in patients with congenital hemophilia," Thank you very much for the suggestions and all the constructive and valuable comments, which have allowed us to improve the quality of the paper. Our point-by-point responses to them are given below. Changes are marked in the revised manuscript using the "Track Changes" function in Microsoft Word.
All the suggestions made by the Reviewers have been implemented in the current version of the paper, and the responses to the comments raised by the Reviewers are attached below.
We are very grateful for all of your comments which have allowed us to improve the quality of the paper.

Review 2:

Comments: I do wonder why the authors used citrullinated histone H3 (CH3) as a marker for osteoporosis? As they mentioned in the discussion, there is thus far only one report pointing to a possible impact of extracellular neutrophil traps on the development of joint injury in hemophilia. However, neither is this marker well approved yet, nor is it associated with bone mineralization or hemophilia in general.

Reply: Thank you for this comment. CH3 served as a marker for the neutrophil trap. Due to the fact that it may have been misrepresented in the text, we have corrected the introduction and emphasized that CH3 is a marker of the extracellular neutrophil trap. The study used osteocalcin (BGLA) as a marker of bone turnover, which is used in medical diagnostics to identify patients with low bone mass. The studies of Christoforidis et al. confirmed the usefulness of BGLA, they reported an increase in BGLA concentration in the serum of patients with hemophilia. Recently, there have also been claims about the possible influence of neutrophilic traps on the development of complications in hemophilia - arthropathy that may be indirectly related to bone loss. Due to the latest reports by Kamiński et al. on the influence of the formation of extracellular neutrophil traps on the body, we decided to include the neutrophil trap marker (citrullinated histone H3 (CH3))

Comments: The authors claim that the combination of PMI, together with laboratory testing of OT and CH3 allows the prediction of low bone mass in patients with hemophilia. I do wonder how the authors are able to draw this conclusion. As they write in the discussion, this may be rather a tool to identify patients with hemophilia, but more - is from my point of view – rather speculative. This is pronounced by the finding, that there is no difference in the concentrations of OT, CH3 and the PMI between the severity of hemophilia groups (mild vs severe). Here, DXA measurements and assessing the bone mineralization status of the study participants is lacking in order to draw any conclusions.

Reply: Thank you for this comment. The PMI is what is known as the bone radiomorphic index. Dental radiology has, for years, investigated the usefulness of the radiomorphic index in identifying patients with low bone mass. There are numerous studies supporting the usefulness of these markers, but they are not related to hemophilia. So far, panoramic x-ray images have been successfully used in identifying patients with low bone mass, primarily postmenopausal women. The use of panoramic x-ray images as a screening tool was based on the particular types of measurements that could be performed in the mandibular region of bone to calculate the so-called radio-morphometric indices. It has been shown that specific indices values correlate with low bone mass and suggest detailed diagnostics for osteoporosis by means of sensitive methods such as DXA. We would like to emphasize that our goal was not to unambiguously diagnose osteoporosis but to identify patients with hemophilia who needed an urgent diagnosis for bone loss, however, we agree with the need to extend the study to DXA results, so we decided to mention this in our conclusions.

Comments: A study from 2015, namely 'The evaluation of MCI, MI, PMI and GT on both genders with different age and dental status' by Bozdag and Sener could show that dental status had a significant effect on the MI and PMI in males. I thus wonder if the authors have assessed the dental status in their study since this could be a confounding factor?

Reply: Thank you for your comment. According to the cited study, oral health, especially tooth loss, may affect MI and PMI. This study, however, is a controversial one, as numerous previous studies have shown that despite the loss of the mandibular alveolar part with tooth loss, the distance of the mental foramen from the lower mandibular border remains relatively constant throughout life

(Wical KE, Swoope CC. Studies of residual ridge resorption. Part II: the relationship of dietary calcium and phosphorus to residual ridge resorption. J Prosthet Dent 1974;32:13-2

Gabriel AC. Some anatomical features of the mandible. J Anat 1958;92:580-6.

Tebo HG, Telford IB. An analysis of the variations in position of the mental foramen. Anat Ret 1950;107:61-6.

Morant GM. A biometric study of the human mandible. Biometrika 1936;28:84- 112).

These findings allowed for the determination and application of bone radiomorphs to predict the risk of reduced bone mass. The assumption, based on one study, that MI and PMI depend on the condition of the oral cavity and tooth loss refutes the validity of determining bone radiomorphic indices. At the same time, according to the authors, it makes the quoted study controversial. For this reason, we decided not to include it in our discussion. Moreover, in our study we showed no statistically significant difference in the MI index values ​​between the study group and the control group. On this basis, and on the basis of the research cited in the article, we conclude that a statistically significant difference in the PMI value was highly likely to be related to the difference in bone mass. Changes in the concentration of osteocalcin seem to confirm our assumptions. Of course, the research should extend to DXA - which we hope for in the future.

At the same time, we agree with the statement that oral health may be associated with osteoporosis. This is mentioned, inter alia, by Kenneth E. Wical et al. In the article "Studies of residual ridge resorption. II. The relationship of dietary calcium and phosphorus to residual ridge resorption.". The authors note, however, that osteoporosis may be related to the periodontium and not to the disease of the hard tissues of the tooth. Therefore, prior to this study, we assessed the periodontium in the study group and in the control group (we assessed the Clinical Attachment Level – CAL in sextants and periodontal pockets (PD) for each tooth present in the oral cavity). We did not observe any significant changes in the periodontium in the study group, which allowed us to proceed with this study. We showed differences in the health of the hard tissues of the teeth (DMFT, DMFS indicators), but on the basis of the above-mentioned studies and our knowledge, we have no grounds to believe that dental caries will be related to the value of PMI, MI indicators or the development of osteoporosis. Oral health in patients with congenital hemophilia was reported in another of our studies that was accepted for publication (https://doi.org/10.1055/s-0042-1743156.). Nevertheless, we are pleased with this question, as it provokes discussion and reflection.

Comments: Why do the authors use different numbers although having the same affiliation (2-4)?

Reply: Thank you for this comment. We have used other affiliate numbers to provide an e-mail address for possible correspondence. As suggested, in order to make the article transparent, we have adjusted the affiliation.

Comments: The authors use twice the spelling haemophilia instead of hemophilia (in the Abstract and in Section 2.4). Please change this in order to be consistent. The same is true for Hematology/Haematology.

Reply: Thank you for this important comment. We have corrected the manuscript so as not to confuse the British and American versions.

Comments: In the Abstract several space characters are missing, e.g. lane 25: …ELISA testsand panoramic…, lane 30: …CH3.The ROC…, lane 34: …who requirea detailed.. Simultaneously, there may be some space characters too much (lane 32, 33).

Reply: Thank you. The problem with spaces is caused by the different versions of MS Word used by individual authors. We have manually corrected the space issue and hope it will not appear anymore.

Comments: Page 2, lane 50: …for most patients, particularly FOR those with hemophilia…

Reply: Thank you. We have corrected the error.

Comments: Page 2, lane 87: panoramic x-ray images and not x-rays images

Reply: Thank you. We have corrected the error.

Comments: Page 2, lane 89: there might be a space character too much between Vistapano and S

Reply: Thank you. We have corrected the error.

Comments: Page 2, lane 91: there might be a space character too much after: The x-rays were…

Reply: Thank you. We have corrected the error. The problem with spaces is caused by the different versions of MS Word used by individual authors. We have manually corrected the space issue and hope it will not appear anymore.

Comments: Page 2, lane 92/93: the authors mixed up the abbreviations for MI and PMI, respectively.

Reply: Thank you for your attention. An error did indeed creep in. We have now improved the expansion of shortcuts.

Comments: Page 2, lane 93: please add the according reference number after Benson et al.

Reply: Thank you. We have added a reference number.

Benson BW, Prihoda TJ, Glass BJ. Variations in adult cortical bone mass as measured by a panoramic mandibular index. Oral Surg Oral Med Oral Pathol. 1991, 71, 349–56. https://doi.org/10.1016/0030-4220(91)90314-3

Comments: Page 3: lane 99: there might be a space character too much after the point.

Reply: Thank you. We have corrected the error.

Comments: Page 3: lane 107: there might be a space character too much after Cropped.

Reply: Thank you. We have corrected the error.

Comments: Page 4: lane 127: please write 25th instead of 25.

Reply: Thank you. We have corrected the error.

Comments: Page 4: lane 129: it is called Mann-Whitney U test.

Reply: Thank you for your comment. We have corrected the name of the test.

Comments: Table 1: there might by a space character too much before 'Type of hemophilia'

Reply: Thank you. We have corrected the error.

Comments: Table 1: I guess 'The Control Group' should be written at the same height as 'The Study Group'

Reply: The style and format of table 1 has been updated and simplified

Comments: Page 4: lane 143: the authors write PMI concentrations; please change the wording.

Reply: Thank you for your comment. We have corrected the sentence. There were statistically significant differences in BGLAP concentration, CH3 concentrations, and PMI values between the study group and the control group.

Comments: Page 4: lane 145: please only write PMI and not PMI index.

Reply: Thank you. We have corrected the error.

Comments: Page 4: Section 3.2.: At first, the authors write: No correlation was found for the PMI index in particular subgroups of patients divided into type, severity, or type of therapy. However, two sentences later they write: There was no correlation between the severity of hemophilia… and the PMI value. Isn't this the same?

Reply: Thank you for your suggestion. The information has indeed been duplicated. This is a deliberate procedure to emphasize in one sentence the lack of correlation between the individual features related to hemophilia (severity, type, type of therapy). The next sentence was to emphasize the lack of coloration of the severity of hemophilia and other variables tested (BGLAP, CH3, and PMI).

Comments: Page 4: lane 150: Until here, nowhere was mentioned, that the occurrence of arthropathy was also assessed.

Reply: Thank you for this comment. We have added a new point in the methodology.

2.4. Medical documentation

The available medical records provide information on the type of hemophilia (A, B), the form of hemophilia (mild, moderate, severe), the treatment of hemophilia, the onset of hemophilic arthropathy and the patient's age.

Comments: Table 2: It should be 'citrullinated histone H3 (CH3)' instead of 'citrullinated histone CH3'.

Reply: Thank you. We have corrected the error.

Comments: Table 2: PMI already stands for panoramic mandibular index, so please write PMI instead of PMI index.

Reply: Thank you. We have corrected the error.

Comments: Table 2: It is called Mann-Whitney U test and not Mann-Whitney test.

Reply: Thank you. We have corrected the error.

Comments: Figure 6: After the point a space character is missing.

Reply: Thank you. We have corrected the error.

Comments: Table 3: Please list the abbreviations used in the legend.

Reply: Thank you for your suggestion. We have corrected the previously unexplained abbreviation. SE - standard error.

Comments: Table 3: Please check the β value for the intercept.

Reply: Thank you. The value provided is correct. Due to the width of the table, some of the values were set in the lower line. We have improved the width of the table.

Comments: Figure 7: for me the caption is not really clear: for what stands the 're'? Please also provide information for the abbreviations to the reader.

Reply: Thank you. We developed the shortcuts. AUC – Area Under Curve, SE – standard error. "Re" is a so-called typo that was created when editing the draft. We have corrected the error.

Comments: Page 8, lane 208: Please add reference.

Reply: Thank you. We have added the reference.

Comments: Thank you for this suggestion. In line with the reviewers' comments, the discussion was corrected.

Reply:

Comments: The section 'Author Contributions' should be over-worked.

Reply: Thank you. The Author Contributions section has been modified and updated

Comments: Section 'Data Availability Statement': Sentence written in double; delete one.

Reply: Thank you. We have removed the duplicate name.

Reviewer 3 Report

Dear Authors

I read with interest your manuscript. It has innovation regarding assessment bone mineral density using mandible. However, the following concerns need to be addressed:

Abstract:

  1. Please include type and severity of hemophilia in the abstract.
  2. Type setting needs to be checked for errors such as: “ with ELISA testsand panoramic x-rays were obtained” line 25. Punctuation needs to be revised since there are multiple punctuation errors such as “CH3.The ROC” and “control group (p<0.05). Multivariate logistic” in line 29.

Introduction:

  1. The sentence: “Techniques for diagnosing osteoporosis, such as Dual-energy X-ray absorptiometry 43 (DXA), are limited and are not routinely used to detect the disease in patients.” Cannot be true, since DEXA is common technique which is used routinely and daily by rheumatologists and endocrinologist for detection reduced bone density.
  2. The authors did not complete literature review in the field of hemophilia and reduced bone density. They have missed earlier papers such as:
  3. 1111/j.1756-185X.2009.01394.x.
  4. 1007/s00296-008-0591-y.
  5. 1016/j.blre.2018.05.002.
  6. 11005/jbm.2017.24.4.201.
  7. 1542/peds.114.2.e177.

These articles should be part of literature review.

Materials and Methods:  

  1. The authors have adjusted control group based on taking history for steroid therapy, chronic renal failure, prolonged immobilization, parathyroid, and thyroid disorders. According to similar survey such evaluation should be associated with conducting some medical laboratory test to can roll out such disease. Asking a question from a patient cannot be sufficient due to some individuals have the disorders but they do not know.
  2. Also health control should be matched regarding age, sex, nationality, height and weight. This is mandatory to have similar bone structure between individuals.

  1. The author said: The detailed data about the study and the control group are presented in Table 1. There are no detailed data about control group in this table.

  1. Based on pervious studies, it was needed to measure blood levels of serum calcium, phosphorus, alkaline phosphates (ALP), serum glutamic oxaloacetic transaminase (SGOT) and serum glutamic pyruvic transaminase (SGPT) and testostrone level, while it is missed in this study.

  1. It has been observed that higher serum osteocalcin levels are relatively well correlated with increases in bone mineral density. While it has earlier been showed patients with hemophilia have lower osteocalcin level compared to contro group (Acta Med Iran 2018;56(3):166-169.), this study shows higher osteocalcin level that is not compatible with reduced bone density in hemohilia.

Radiological study protocol:

  1. Usage of abbreviations of “MI (panoramic mandibular in-92 dex) and PMI (Mental Index)” is not correcte and needs to be replaced.
  2. What does “ten” mean in line 100?
  3. In footnote of figures 1 and 2. Please indicate that these pictures are from patients or control group.

Author Response

We wish to resend our review article, "Potential biochemical markers and radio-morphometric indices as predictors of reduced bone mass in patients with congenital hemophilia," Thank you very much for the suggestions and all the constructive and valuable comments, which have allowed us to improve the quality of the paper. Our point-by-point responses to them are given below. Changes are marked in the revised manuscript using the "Track Changes" function in Microsoft Word.
All the suggestions made by the Reviewers have been implemented in the current version of the paper, and the responses to the comments raised by the Reviewers are attached below.
We are very grateful for all of your comments which have allowed us to improve the quality of the paper.

Review 3:

Comments: Abstract:

Please include type and severity of hemophilia in the abstract.

Reply: Thank you. We have revised the abstract. The study consisted of 50 adult men with hemophilia A and B (mild, moderate, severe).

Comments: Type setting needs to be checked for errors such as: "with ELISA tests and panoramic x-rays were obtained" line 25.

Reply: Thank you. We have corrected the error.

Comments: Punctuation needs to be revised since there are multiple punctuation errors such as "CH3.The ROC" and "control group (p<0.05). Multivariate logistic" in line 29.

Reply: Thank you. We have corrected the error. Due to different versions of MS Word, certain authors had punctuation errors that we tried to correct manually.

Comments: Introduction:

The sentence: "Techniques for diagnosing osteoporosis, such as Dual-energy X-ray absorptiometry 43 (DXA), are limited and are not routinely used to detect the disease in patients." Cannot be true, since DEXA is common technique which is used routinely and daily by rheumatologists and endocrinologist for detection reduced bone density.

Reply: Thank you for your suggestion. Of course, we are aware that there are medical fields such as rheumatology and endocrinology that are based on DXA. In our introduction, we meant that DXA is not screened in all hemophilia patients (by hematologists), despite the risk of reduced bone loss. We have now corrected this sentence in the article.

Techniques for diagnosing osteoporosis, such as Dual-energy X-ray absorptiometry (DXA), are limited and are not routinely used to detect the disease in patients with hemophilia.

Comments: The authors did not complete literature review in the field of hemophilia and reduced bone density. They have missed earlier papers such as:

1111/j.1756-185X.2009.01394.x.

1007/s00296-008-0591-y.

1016/j.blre.2018.05.002.

11005/jbm.2017.24.4.201.

1542/peds.114.2.e177.

These articles should be part of literature review.

Reply: Thank you for your suggestions and valuable literature. We have expanded our article with the mentioned items Thank you for this suggestion. In line with the reviewers' comments, the discussion was corrected.

Comments: Materials and Methods: 

The authors have adjusted control group based on taking history for steroid therapy, chronic renal failure, prolonged immobilization, parathyroid, and thyroid disorders. According to similar survey such evaluation should be associated with conducting some medical laboratory test to can roll out such disease. Asking a question from a patient cannot be sufficient due to some individuals have the disorders but they do not know.

Also health control should be matched regarding age, sex, nationality, height and weight. This is mandatory to have similar bone structure between individuals.

Reply: Thank you for this comment. We tried to match the test and control groups to have a similar bone structure between individuals. For this reason, the control group is men (hemophilia A and B occur mainly in men, which is related to the inheritance syndrome). In addition, we collected a questionnaire from patients containing, inter alia, sociodemographic data. In order to improve the adjustment of the control group to the study group, we decided to disqualify patients from the study group who stated that they were aware of the presence of potential comorbidities that could affect the reduced bone mass (e.g. chronic steroid therapy). We are aware that the patient may be unaware of many diseases but it is not possible to perform all tests on all patients. We are aware of the limitations of the study, therefore, as suggested, we have added information in the article on this topic. An unquestionable limitation of our study are the financial limitations, which made it impossible to correlate the obtained measurements with DXA. However, taking into account the fact that the center where the research was conducted is the only center in the voivodeship that implements the "National Program of Treatment of Patients with Hemophilia and Related Hemorrhagic Diathesis" for adult patients, there is hope in the future to extend the research.

Comments: The author said: The detailed data about the study and the control group are presented in Table 1. There are no detailed data about control group in this table.

Reply: Thank you for your suggestion. We have corrected the sentence. The data about the study and the control group are presented in Table 1. There are no detailed data about the control group in this table. The table contains detailed information about the study group and the most important information about the control group.

Comments: Based on previous studies, it was needed to measure blood levels of serum calcium, phosphorus, alkaline phosphates (ALP), serum glutamic oxaloacetic transaminase (SGOT) and serum glutamic pyruvic transaminase (SGPT) and testostrone level, while it is missed in this study.

Reply: Thank you for your suggestions. Unfortunately, due to financial constraints, we were unable to measure all the markers we had planned. We hope that in the future it will be possible to obtain funding that will allow us or other researchers to expand the study. As with any study, our study has some limitations, but nevertheless, we believe that our study has shown potential new lines of research that we hope will be able to continue. We used an extracellular neutrophil trap marker in the study, which so far has been poorly studied, especially in the case of hemophilia, which is a great advantage of the work. Extracellular neutrophil traps are a new and continuously studied mechanism by which neutrophils fight infection. However, recent literature shows the negative effects of neutrophil trap formation in various systemic diseases. So far, we have only found one study suggesting a negative effect of neutrophil trap formation on the development of hemophilic arthropathy. Our study also shows a possible link between NETs and the development of complications in hemophilia, which may inspire further researchers and thus lead to new discoveries. In addition, our study indicated a possible use of routinely performed orthopnea tomographic examinations. This is a method known in dental radiology for years, but is not used in other fields of medicine. For this reason, we believe that it is worth extending this research and paying attention to the possible use of dental radiographs, not only for the needs of dentists but also hematologists.

Comments: It has been observed that higher serum osteocalcin levels are relatively well correlated with increases in bone mineral density. While it has earlier been showed patients with hemophilia have lower osteocalcin level compared to control group (Acta Med Iran 2018;56(3):166-169.), this study shows higher osteocalcin level that is not compatible with reduced bone density in hemophilia.

Reply: Thank you for this comment and for the valuable exchange of information. We agree that there are studies that show that hemophilia patients may have lower levels of osteocalcin. As we have shown in the discussion, there are also numerous studies which, like ours, have shown a higher concentration of osteocalcin in the plasma of patients with congenital hemophilia. We agree that for this reason it is necessary to continue research and share the obtained results. According to the authors, it is worth considering what the causes of the discrepancies in the obtained results are. Hemophilia is a genetic disease and is relatively rare. In our study, we managed to examine a significant proportion of patients with congenital hemophilia in the Wielkopolskie voivodeship (Poland) - the only center in the voivodeship that implements the "National Program for the Treatment of Patients with Hemophilia and Related Hemorrhagic Disease" for adult patients. Perhaps the discrepancies between researchers from different parts of the world result from an unknown / so far unexplored mutation? Bone remodeling is a dynamic process involving bone formation and resorption. According to the authors, the higher concentration of osteocalcin (and thus the higher activity of osteoblasts) may be due to increased bone remodeling and may be a consequence of bone loss (recompensation).

Comments: Usage of abbreviations of "MI (panoramic mandibular in-92 dex) and PMI (Mental Index)" is not correct and needs to be replaced.

Reply: Thank you for pointing out the error. We have now improved the shortcuts.

Comments: What does "ten" mean in line 100?

Reply: Thank you for pointing out any inaccuracies. This piece of text has been re-edited to avoid confusion.

Comments: In footnote of figures 1 and 2. Please indicate that these pictures are from patients or control group.

Reply: Thank you for your comment. We have supplemented the data with this information.

Thank you again for the review.

Round 2

Reviewer 1 Report

The authors have answered all questions previously done and corrected in the manuscript what was suggested.

Reviewer 3 Report

Dear authors

The article in the current format is more informative and scientific now.